# National Health Insurance Membership among Urban Poor Societies in Indonesia in 2019: Are They Protected?

Agung Dwi Laksono [1], Rukmini Rukmini [1], Tumaji Tumaji [1], Mara Ipa [1] and Ratna Dwi Wulandari [2,*]

1   National Research and Innovation Agency Republic of Indonesia, Jakarta 10340, Indonesia
2   Faculty of Public Health, Universitas Airlangga, Surabaya 60115, Indonesia
*   Correspondence: ratna-d-w@fkm.unair.ac.id

**Abstract:** Urban poor societies were a vulnerable group. Once they are sick, they fall deeper into poverty. National Health Insurance (NHI) is a way the government initiated to overcome this situation. We analyzed the factor related to NHI membership among urban poor societies. The study population included everyone living in urban poor societies. The study examined a sample of 3455 participants, and examined five characteristics: NHI, age, gender, education, employment, and marital status. In the final step, the research used binary logistic regression. The results show that all age groups are more likely than those over 64 to be a member of NHI among urban poor societies in Indonesia. Males have a 1.039 times higher chance than females to be a member of NHI. All education levels have less possibility than no education to be a member of NHI among urban poor societies in Indonesia. Employees have an opportunity of 1.097 times higher than the unemployed to be a member of NHI. All marital statuses have more possibility than those who are divorced or widowed to be a member of NHI among urban poor societies in Indonesia. The study results strengthen previous studies with a similar topic. We concluded that the NHI adequately protects urban poor societies. However, the government still has homework to pay attention to the remaining 28.3% of urban poor societies that the NHI has not covered.

**Keywords:** urban poor society; health insurance; national health insurance; health policy; big data; population survey; public health



## 1. Introduction

Health insurance is a type of insurance product that aims to guarantee the health or treatment costs of the insured (Goretti and Aditya 2019). This health insurance is vital for managing health risks and financial losses and reducing vulnerability to poverty (Zheng and Peng 2021). The development of health insurance in Indonesia is inseparable from the health system, which aims to ensure that all people have access to quality and effective promotive, preventive, curative and rehabilitative health services, through social insurance mechanisms and the principle of equity by requiring every resident who can afford to do so to pay contributions, while those who cannot afford to pay contributions receive assistance from the government (Nasution et al. 2020; Laksono et al. 2021b; Wulandari and Laksono 2021).

The poor are a vulnerable group entitled to special treatment and protection regulated by law (Laksono et al. 2020, 2021c). Meanwhile, the growth of the urban poor relative to the number of poor people is increasing very rapidly. During 1990–2008, poverty in urban populations in Asia increased to 21.9% (Mathur 2013). Likewise, with poverty in urban areas in Indonesia, the percentage of urban poor people in March 2021 rose to 12.18 million, equivalent to 7.89% (Central Bureau of Statistics Republic of Indonesia 2021). Increasingly, the poor in urban areas need protection and social security programs, including health insurance. Forty-eight studies from 17 countries show an average health insurance participation rate of 36% for the vulnerable population. At the same time, households from the wealthiest group have a 61% higher chance of taking insurance than the poorest group. This condition shows



that despite tremendous efforts from the government, health insurance schemes in low- and middle-income countries generally do not reach the targeted vulnerable population (Osei Afriyie et al. 2022).

The Indonesian Government is developing a health insurance policy, namely, by launching a single-payer health insurance program, i.e., the National Health Insurance (NHI), to achieve universal health coverage (UHC) in 2019 (Agustina et al. 2019; Laksono et al. 2021c). This program integrates and replaces all previously fragmented social health. Besides covering the poor and near-poor, the NHI scheme is mandatory for everyone in Indonesia, with different time limits to protect the entire population (Mboi et al. 2018). The Social Security Administering Agency (SSAA) carries out its duties as the organizer of the National Social Security System (NSSS) based on the principles of humanity, benefits, and social justice for all Indonesian people (Nasution et al. 2020). The situation shows that the Indonesian Government has provided health insurance for its people, especially the poor.

However, implementing the NHI program encountered several problems, including the findings regarding the participants in the regional proposals for contribution assistance recipients (CAR) who were not well targeted (Nasution et al. 2020). The low number of NHI participants among informal workers is due to complicated procedures, place of registration, premium payment, and the assumption that there are additional costs, as well as sociodemographic factors and ownership of a history of chronic disease (Intiasari et al. 2015; Putro and Barida 2017).

Problems related to health insurance associated with the poor also occur in several other countries. There are still significant health disparities in India between states, rural and urban areas, and social classes (RamPrakash and Lingam 2021). A study in China said that social insurance coverage increased access to health services and reduced the proportion of out-of-pocket. Still, health costs for catastrophic diseases for the poor remained high (Fang 2019).

The poor are one of the groups targeted in the universal coverage of NHI through the CAR scheme, which is intended to provide health insurance coverage as a social safety net in the form of disability and pension benefits. In reality, the range is uncertain for near-poor households, especially those working in the informal economy. These are the main barriers to achieving the universal coverage that is targeted to be performed in 2019 (Liyanto et al. 2022). In theory, government policies with CAR should be able to protect the urban poor. Therefore, this study is vital to evaluate the effectiveness of the CAR policy. Based on the background narrative, the study aims to analyze the factors related to NHI membership among urban poor societies in Indonesia.

## 2. Methodology

### 2.1. Data Source

The Ministry of Health of the Republic of Indonesia performed the poll nationwide. Secondary data from a previous study were used in this investigation ("Abilities and Willingness to Pay, Fee, and Participant Satisfaction in implementing National Health Insurance in Indonesia in 2019"). Meanwhile, the study's participants were entirely from Indonesia's urban impoverished societies. The survey collected a total of 47,644 respondents. With the sample criteria in this study, the urban poor, the study identified 3455 participants as research responders. Study participants cover 34 provinces in Indonesia with a sampling technique that considers the provincial and district/city levels through stratification and multistage random sampling.

### 2.2. Variables

The dependent variable was membership in the National Health Insurance (NHI). The respondent's presence in the NHI, whether as an individual member, a required member (civil servant, police, army), borne by the corporation, or a recipient of government donation assistance, is considered membership. Non-members and members are the two forms of NHI membership.

Five factors were included as independent variables in the study. The five factors were age, gender, educational level, employment position, and marital status. The age group was divided into three categories in the sample: under 18, 18–64, and over 64. Meanwhile, the study split gender into two categories: male and female. No education, primary, secondary, and higher education were the four levels of education. Furthermore, there are two different forms of employment status: jobless and employed. The study also separated marital status into three groups: never married, married, and divorced/widowed.

### 2.3. Setting

The study was in Indonesia's urban impoverished societies at the national level. In the survey, the researchers employed the provisions of the Indonesian Central Statistics Agency for urban categorization.

The poll used the wealth index methodology to determine wealth status. A weighted average of a family's overall spending generated the wealth index. Meanwhile, the wealth index was calculated using primary household expenditures such as health insurance, food, and lodging. In addition, the income index was divided into five categories: quintile 1 (poorest), quintile 2 (poorer), quintile 3 (middle), quintile 4 (richer), and quintile 5 (richest) (richest) (Wulandari et al. 2019, 2022)—the research includes the poorer and poorest as the poor.

### 2.4. Data Analysis

The Chi-Square test was employed to create a bivariate comparison in the early stages of the sample. Furthermore, a collinearity test was used to guarantee that the independent variables in the final regression model did not have a strong association. In the study's final point, a binary logistic regression was applied. The survey used this test to look into the multivariate association between all independent variables and NHI membership, the dependent variable. The regression equation used in the study is as follows:

$$\ln\left(\frac{\hat{p}}{1 - \hat{p}}\right) = \beta_0 + \beta_1 X \tag{1}$$

ln: Natural Logarithm, where $B_0 + B_1 X$ is an equation commonly known in OLS, while P Accent is the logistic probability obtained by the following formula:

$$\hat{p} = \frac{\exp(\beta_0 + \beta_1 X)}{1 + \exp(\beta_0 + \beta_1 X)} = \frac{e^{\beta_0 + \beta_1 X}}{1 + e^{\beta_0 + \beta_1 X}} \tag{2}$$

The study employed the IBM SPSS 26 application for the entire statistical analysis process.

### 2.5. Ethical Approval

The survey "Abilities and Willingness to Pay, Fee, and Participant Satisfaction in the Implementation of NHI in Indonesia in 2019" received ethical clearance from the National Ethics Committee (Number: LB.0201/2/KE.340/2019). All respondents' identities were deleted from the dataset as part of the study.

Interventionary studies involving animals or humans, and other studies that require ethical approval, must list the authority that provided approval and the corresponding ethical approval code.

## 3. Results

The study results found that 71.7% of urban poor societies are members of the NHI. Next, Table 1 shows the descriptive statistics of NHI membership among urban poor communities in Indonesia.

**Table 1.** Descriptive statistic of NHI membership among urban poor societies in Indonesia in 2019 (n = 3455).

| Characteristics | NHI Membership | | *p*-Value |
| --- | --- | --- | --- |
| | Member (n = 2508) | Non-Member (n = 947) | |
| Age group | | | <0.001 |
| under 18 | 19.1% | 29.2% | |
| 18–64 | 67.5% | 63.0% | |
| over 64 | 13.5% | 7.8% | |
| Gender | | | <0.001 |
| Male | 46.7% | 49.3% | |
| Female | 53.3% | 50.7% | |
| Education level | | | <0.001 |
| No education | 11.9% | 17.8% | |
| Primary | 58.5% | 55.2% | |
| Secondary | 26.2% | 24.0% | |
| Higher | 3.4% | 3.0% | |
| Employment status | | | <0.001 |
| Unemployed | 52.2% | 55.5% | |
| Formal | 47.8% | 44.5% | |
| Marital status | | | <0.001 |
| Never married | 35.3% | 43.5% | |
| Married | 50.3% | 46.5% | |
| Divorced/Widowed | 14.4% | 9.9% | |

Based on age group, 18–64 led all NHI membership groups. Meanwhile, those who were female and with primary education ruled in both NHI membership categories. Regarding employment status, the unemployed dominated all NHI membership groups. Moreover, based on marital status, married people led both NHI membership groups.

The next step was a collinearity test. The analysis findings show that the independent variables do not have a significant relationship. The tolerance value is more than 0.10 for all variables, and the variance inflation factor (VIF) value is less than 10.00 for all factors simultaneously. The study then stated that the regression model had no symptoms of multicollinearity, pointing to the test's decision-making base.

Table 2 shows the binary logistic regression of NHI membership among urban poor societies in Indonesia. In the last stage, the study used "non-member" as a reference. Based on age groups, Table 2 indicates that under 18 are 2.662 times more likely to be a member of NHI than those over 64 (AOR 2.662; 95% CI 2.647–2.677). Moreover, 18–64 have a possibility of 1.620 compared to those over 64 to be a member of NHI (AOR 1.620; 95% CI 1.613–1.626). The analysis results show that all age groups are more likely than those over 64 to be a member of NHI among urban poor societies in Indonesia. Table 2 shows that males have a chance of 1.039 times higher than females to be a member of NHI (AOR 1.039; 95% CI 1.036–1.041). The result indicates that females are less likely than males to join NHI.

Regarding education level, those with primary education are 0.737 less likely to be a member of NHI than those with no education (AOR 0.737; 95% CI 0.734–0.739). Secondary education is 0.759 less likely than those without education to be a member of NHI (AOR 0.759; 0.756–0.762). Furthermore, higher education is 0.723 less likely to be a member of NHI than those without education (AOR 0.723; 95% CI 0.718–0.728). The results indicate that all education levels have less possibility than no education to be a member of NHI among urban poor societies in Indonesia.

According to the employment status, the employed have a possibility 1.097 times higher than the unemployed to be a member of NHI (AOR 1.097; 95% CI 1.094–1.100). The result means that employment is a protective factor to being a member of NHI among urban poor societies in Indonesia.

Table [2] indicates that those who have never married have a possibility 1.102 times more likely than those who are divorced or widowed to be a member of NHI (AOR 1.102; 95% CI 1.096–1.107). On the other hand, those who married are likely 1.216 times higher than those who divorced or widowed to be a member of NHI (AOR 1.216; 95% CI 1.211–1.221). It indicates that all marital statuses have more possibility than those who are divorced or widowed to be a member of NHI among urban poor societies in Indonesia.

**Table 2.** The result of binary logistic regression of NHI membership among urban poor societies in Indonesia (n = 3455).

| Predictor | *p*-Value | AOR | 95% CI | |
|---|---|---|---|---|
| | | | Lower Bound | Upper Bound |
| Age group: under 18 | <0.001 | 2.662 | 2.647 | 2.677 |
| Age group: 18–64 | <0.001 | 1.620 | 1.613 | 1.626 |
| Age group: over 64 | - | - | - | - |
| Gender: Male | <0.001 | 1.039 | 1.036 | 1.041 |
| Gender: Female | - | - | - | - |
| Education: No Education | - | - | - | - |
| Education: Primary | <0.001 | 0.737 | 0.734 | 0.739 |
| Education: Secondary | <0.001 | 0.759 | 0.756 | 0.762 |
| Education: Higher | <0.001 | 0.723 | 0.718 | 0.728 |
| Employment: Unemployed | - | - | - | - |
| Employment: Employed | <0.001 | 1.097 | 1.094 | 1.100 |
| Marital: Never married | <0.001 | 1.102 | 1.096 | 1.107 |
| Marital: Married | <0.001 | 1.216 | 1.211 | 1.221 |
| Marital: Divorced/Widowed | - | - | - | - |

Note: AOR: Adjusted Odds Ratio; CI: confidence interval.

## 4. Discussion

Of the approximately 270 million Indonesians, currently, 86% are registered as NHI participants. This number includes the poor and underprivileged whose premium contributions are financed by the government as Contribution Assistance Recipients (CAR) (Laksono et al. 2021a). Data up to November 2021 show that the number of active NHI CAR participants reached 88.7 million or 91.63% of the total CAR quota in 2021, as many as 96.8 million (Ardianto 2022). The situation means that there are still around 8 million poor and underprivileged people who have entered the quota as CAR participants but have not been actively registered in the NHI. On the other hand, it is not uncommon in the field to find underprivileged people who have not or are not registered in the CAR quota (Nasution et al. 2019; Andayani et al. 2021).

The results show that all age groups are more likely than those over 64 to be a member of NHI among urban poor societies in Indonesia. This result is in contrast to the results of previous studies, which stated that older people are more likely to participate in health insurance programs with the assumption that they have a greater risk of disease, so they are encouraged to have health insurance (Aregbeshola and Khan 2018; Nguyen and Giang 2021). In addition, getting older is also associated with increasing financial ability, which can buy health insurance policies (Akiyama et al. 2018; Wulandari and Laksono 2019; Nguyen and Giang 2021). A plausible explanation of our findings is that this study was conducted in poor communities where many of the aged no longer have an income or where there may not be income. Their income is much reduced, reducing the possibility of becoming a health insurance participant (Liyanto et al. 2022). With these conditions, they voluntarily or are forced to use life experiences to care for or treat themselves when sick (Atnafu and Gebremedhin 2020).

The results indicate that females are less likely than males to be a member of NHI. This result is in line with a study in Kenya that stated that men were more likely to participate in the National Health Insurance program than women (Kazungu and Barasa 2017). The

condition can happen due to gender barriers, financial dependence, or limited mobility for women (Wulandari et al. 2020a; RamPrakash and Lingam 2021). Another reason is that most women do not have insufficient income and therefore cannot register and pay the National Health Insurance premium (Aregbeshola and Khan 2018).

The study shows that all education levels have less possibility than no education to be a member of NHI among urban poor societies in Indonesia. This result contradicts previous research, which states that higher education is positively related to insurance ownership (Dawit et al. 2020; Laksono et al. 2021a, 2021b; Wang et al. 2021b). A high level of education is assumed to be a better level of knowledge, including knowledge of the principles of the health insurance program and the benefits package, so it is positively correlated with health insurance participation (Aregbeshola and Khan 2018). However, our findings can be justified in the context of the poor in Indonesia. In general, the level of education is positively correlated with the wealth or economic status of the community—the lower the education level, the lower the economic level and vice versa (Wulandari et al. 2020b; Liu et al. 2021).

In the National Health Insurance program, as stated in law number 40 of 2004, number 14, it is stated that the poor are registered in the National Health Insurance membership as PBI participants. The situation gives the uneducated people who are the poor a greater chance of participating in the National Health Insurance program than people with higher education. In other cases, the level of education is not directly proportional to the ownership of health insurance. Other factors are considered more important than affecting people's ownership of health insurance, one of which is experience. A study in India shows that past illness experiences significantly influence health insurance ownership more than education (Savitha and Banerjee 2021). Likewise, the research results in Bangladesh and Nepal show that one factor influencing households to register for the National Health Insurance program is a history of chronic disease in the family (Mahmood et al. 2018; Ghimire et al. 2019).

The result shows that employment is related to the NHI membership among urban poor societies in Indonesia. The condition is undoubtedly not surprising because working people earn money, which can be allocated partly to pay for health insurance premiums. These findings align with previous research, stating that working people are more likely to participate in health insurance (Mahmood et al. 2018; Weldesenbet et al. 2021). On the other hand, regarding people who do not work and have financial limitations, this causes them not to register for national health insurance. A lack of financial resources became a barrier for people to register for health insurance (Laksono et al. 2021a). Lack of income causes people to feel unable to pay insurance premiums, so they are reluctant to become participants of the national health insurance. However, health insurance premiums that must be paid regularly become a burden for those who do not have jobs (Nguyen and Giang 2021).

It indicates that all marital statuses have more possibility than those who are divorced or widowed to be a member of NHI among urban poor societies in Indonesia. The situation is in line with the results of previous studies, which state that married people are more likely to become participants in the national health insurance than those who are single or divorced (Kazungu and Barasa 2017). The condition is probably due to the level of financial solvency among married people (Liu et al. 2021; Megatsari et al. 2021). Married people get financial support from their families or at least get encouragement from their partners to enable them to participate in the national health insurance (Mulenga et al. 2017). Meanwhile, our findings also show that unmarried people are more likely to have health insurance than divorced people. It may be because those who are not married have no dependent children, so it is easier to pay insurance premiums.

In general, the results of this study strengthen the results of previous studies on health insurance, both in the context of targeting the general public and with more specific targets. However, the Indonesian government still has homework to pay attention to the remaining 28.3% of urban poor societies that the NHI has not covered. The government still has to

release a policy that can increase NHI participation as mandated by law. According to the study's results, the target for the urban poor is those over 64, female, with better education, unemployed, and divorced or widowed.

*Study Limitation*

This study has advantages because it uses big data as an analysis material, allowing the findings to be extrapolated to the national level. The analysis in this study, on the other hand, is based on secondary data. The acceptable variables are those provided by the Ministry of Health of the Republic of Indonesia. Several other factors that were previously known to influence health insurance ownership could not be investigated. These characteristics include cognitive ability, prior commercial insurance ownership, having children, and family size (McGarry et al. 2018; Alo et al. 2020; Wang et al. 2021a).

**5. Conclusions**

The study concluded that the NHI adequately protects urban poor societies. Moreover, the five variables tested have a relationship with NHI membership, and the five were age group, gender, educational level, employment, and marital status. Moreover, the Indonesian government still has work to do to address the remaining 28.3 percent of urban poor societies not covered by the NHI. As required by law, the government still must issue a policy to increase NHI participation.

**Author Contributions:** Conceptualization, A.D.L. and R.D.W.; methodology, A.D.L.; software, R.R. and T.T.; validation, A.D.L. and M.I.; formal analysis, R.R. and M.I.; investigation, R.R., T.T. and M.I.; resources, R.D.W.; data curation, T.T.; writing—original draft preparation, R.R., T.T., M.I. and R.D.W.; writing—review and editing, A.D.L.; visualization, T.T.; supervision, A.D.L.; project administration, M.I.; funding acquisition, A.D.L. All authors have read and agreed to the published version of the manuscript.

**Funding:** The author received no funding for this study.

**Institutional Review Board Statement:** The 2017 IDHS uses the Standard DHS survey technique, initially reviewed and approved by the ORC Macro IRB (Institutional Review Board) in 2002 as part of the Demographic and Health Surveys (DHS) Program (DHS-7) authorized by ICF International's Institutional Review Board. DHS surveys that satisfy the requirements are branded as DHS-7 program approved, and the required paperwork is provided. ICF International's Institutional Review Board adheres to the US Department of Health and Human Services' "Protection of Human Subjects" requirements (45 CFR 46).

**Informed Consent Statement:** Respondents have provided written consent for their involvement in the 2017 IDHS.

**Data Availability Statement:** The authors cannot publicly share the data because a third party and authors do not have permission to share it. The 2017 IDHS data set name requested from the ICF 'data set of childbearing age women' is available from the ICF contact https://dhsprogram.com/data/new-userregistration.cfm (accessed on 18 June 2022) for researchers who meet the criteria for access to confidential data.

**Conflicts of Interest:** The authors declare no conflict of interest.

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
