# Peer review of "National Health Insurance Membership among Urban Poor Societies in Indonesia in 2019: Are They Protected?"

_economies, doi:10.3390/economies10080196_

Round 1

Reviewer 1 Report

The authors use logistic regression to identify the factors that determine access to NHI for the poorest part of the population. The topic is interesting, but the paper in some of its parts is not very clear.

I would suggest clarifying the research steps listed here:

1. The relationship between poverty and NHI is only made explicit in the literature review, but is not used in the model. In this way, the abstract and introduction seem detached from the rest of the paper and it becomes difficult to understand the purpose of the research on first reading. The abstract and introduction should be made consistent with the rest of the paper. As interesting as it is, the relationship between NHI and the risk of poverty is not analyzed, since a sub-sample of people who are already poor are analyzed;

2. In the description of the data, it is unclear whether the sample of 3,455 participants relates to the total respondents of the Ministry of Health of the Republic of Indonesia survey or the subsample of the two poorest quantiles actually used in the analysis. This would also make the reading of Table 1 clearer;

3. What are the hypotheses of the Chi-square test presented in paragraph 2.4? Is it the same as table 1?

4. Some counterintuitive results (e.g. odds ratios for education) are justified by arguing that there is self-selection of the sample due to the selection of the poorest subsample. This could be demonstrated by comparing the results of an estimated model on the remainder of the population or by introducing the quantiles of income as a response variable in a model on the whole population.

Author Response

The authors use logistic regression to identify the factors that determine access to NHI for the poorest part of the population. The topic is interesting, but the paper in some of its parts is not very clear.

I would suggest clarifying the research steps listed here:

  1. The relationship between poverty and NHI is only made explicit in the literature review, but is not used in the model. In this way, the abstract and introduction seem detached from the rest of the paper and it becomes difficult to understand the purpose of the research on first reading. The abstract and introduction should be made consistent with the rest of the paper. As interesting as it is, the relationship between NHI and the risk of poverty is not analyzed, since a sub-sample of people who are already poor are analyzed;

Response: The population in this study is the poor. Of course, this article does not discuss the relationship between poverty and NHI.

  1. In the description of the data, it is unclear whether the sample of 3,455 participants relates to the total respondents of the Ministry of Health of the Republic of Indonesia survey or the subsample of the two poorest quantiles actually used in the analysis. This would also make the reading of Table 1 clearer;

Response: the description was revised as suggested.

  1. What are the hypotheses of the Chi-square test presented in paragraph 2.4? Is it the same as table 1?

Response: Yes, it is. Table 1 is the result of the chi-square test

  1. Some counterintuitive results (e.g. odds ratios for education) are justified by arguing that there is self-selection of the sample due to the selection of the poorest subsample. This could be demonstrated by comparing the results of an estimated model on the remainder of the population or by introducing the quantiles of income as a response variable in a model on the whole population.

Response: However, our findings can be justified in the context of the poor in Indonesia. In general, the level of education is positively correlated with the wealth or economic status of the community—the lower the education level, the lower the economic level and vice versa (Wulandari, Putri and Laksono, 2020; Liu et al., 2021), and the poor are targets of the Indonesian government's CAR policy to overcome the health financing barrier. This finding actually shows the effectiveness of the CAR policy released by the Indonesian government.

Reviewer 2 Report

We appreciate this study focusing on the strategic issue in Indonesia, the aspects of National Health Insurance (NHI).  

The manuscript presents the results of the analysis, which provide a novelty as the core of the scientific article. However, problems appear in the paper, such as in the abstract, introduction, methods, discussion, conclusion, and references. For more detail, I mention comments as attached follows.

Author Response

We appreciate this study focusing on the strategic issue in Indonesia, the aspects of National Health Insurance (NHI).  

The manuscript presents the results of the analysis, which provide a novelty as the core of the scientific article. However, problems appear in the paper, such as in the abstract, introduction, methods, discussion, conclusion, and references. For more detail, I mention comments as attached follows.

Abstract

  1. The abstract should not contain statistical indicators

Response: the statistical indicators were deleted from the abstract.

  1. This section should present the contribution of the findings to the economic theory and government policy.

Response: the suggestion was added to the abstract.

Introduction

The introduction should present the target of the scientific contribution at the end of this section.

Response: the suggestion was added to the introduction.

Research Methods

  1. Please explain the spatial aspect/location aspect of the respondents

Response: the suggestion was added to the research methods.

  1. This section should express the regression equations used in the analysis.

Response: the regression equations was added to the research methods.

Results and discussion

  1. Please attach the empirical estimation results (binary regression) summarized in table 2 in the appendix.

Response: the author added the empirical estimation results (binary regression) in the appendix.

  1. Some paragraphs contain less than three sentences; combine them into one section.

Response: the paragraphs were combined as suggested.

  1. This section contains the research report, interpretation, and discussion. However, this section has not highlighted scientific contributions yet. The authors should declare the scientific contribution of the research findings to economic theory and policy implications.

Response: the discussion was revised as suggested.

Conclusion

This section contains limited sentences. It may present novelty and policy implications.

Response: the conclusion was revised as suggested.

References

It contains two references in the Indonesian language. Try to replace it with another in English.

Response: the references were revised as suggested

Round 2

Reviewer 1 Report

Dear authors,

thank you to take into consideration my suggestions.

The manuscript is suitable for pubblication.